# FlockLab 2: Multi-Modal Testing and Validation for Wireless IoT

Roman Trüb, Reto Da Forno, Lukas Sigrist, Lorin Mühlebach, Andreas Biri, Jan Beutel,
Lothar Thiele
Computer Engineering and Networks Laboratory, ETH Zurich
Switzerland
rtrueb@ethz.ch

## ABSTRACT

The development, evaluation, and comparison of wireless IoT and cyber-physical systems requires testbeds supporting inspection of logical states and accurate observations of physical performance metrics. We present FlockLab 2, a second generation testbed supporting multi-modal, high-accuracy and high-dynamic range measurements of power and logic timing and at the same time in-situ debug and trace infrastructure of modern microcontrollers allowing for reproducible evaluation and benchmarking. We detail the architecture, provide a characterization and demonstrate the interface, the supported services and the tools of the FlockLab 2 testbed.

*Data Availability Statement.* The hardware design and the software for server and observer of the presented testbed architecture and the data for the plots in this paper are openly available at https://flocklab.ethz.ch.

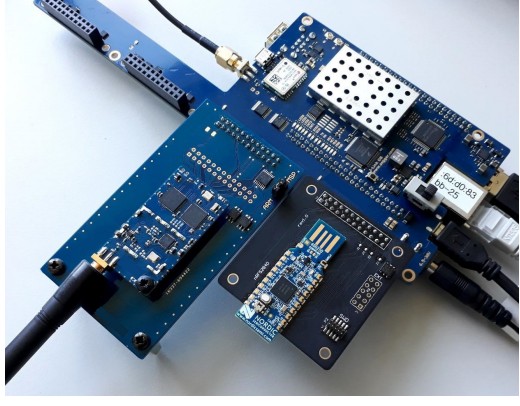

**Figure 1: FlockLab 2 observer with 4 target slots.**

## 1 INTRODUCTION

The ever-increasing complexity and care for detail that must be mastered in developing state-of-the-art distributed networked embedded applications requires modern and adequate tool support for experimentation. In scaling to large distributed applications, simulations can help but cannot replace experiments on real hardware. Simulation always implies simplifications, significant especially at the hardware level. The latest microcontrollers and radios used in wireless Internet of Things (IoT) applications feature numerous power modes that need to be accurately fine-tuned and orchestrated for efficiency. The interaction between peripherals and the system core needs to be well-understood and validated for reliable operation down to the instruction level. Timing needs to be controlled at application as well as driver level up to the speed of light, as recent work on network protocols incorporating the time-of-flight of radio signals has shown [10].

The development of embedded software is commonly based on state-of-the-art debug and trace infrastructure integrated into the hardware of modern microcontroller architectures [15]. This in-situ infrastructure is supported by a multitude of development tools that can be used on the user's desk and also remotely. Today, such tooling is limited to a single device-under-test (DUT), therefore severely limiting capabilities to develop and test algorithms and systems for distributed wireless IoT devices. It is exactly this distributed nature of many devices, coupled over variable wireless channels and directly influenced by the embedding environment, that is known to be a challenging task in designing, implementing and validating IoT and cyber-physical systems. Therefore, distributed testbeds with representative hardware deployed in a real environment are widely used. Such testbeds allow (1) the reuse of the testing infrastructure, (2) controlled and reproducible testing and validation, and (3) the comparison of different implementations on a common platform (benchmarking). A number of testbeds exist that support a subset of the aspects mentioned above that are required for contemporary software development and evaluation for wireless IoT devices. An overview of existing testbeds and their capabilities as well as our remarks on 8+ years of testbed development and operation is provided in Sec. 2. However, none of the existing testbeds supports combined native in-situ debug and trace infrastructure, accurate timing measurements as well as the detailed assessment of power consumption over a large dynamic range.

In this work, we present a versatile testbed with capabilities addressing the aforementioned requirements. In jointly addressing challenges in power measurement, timing, functional correctness based on native, in-hardware debug and trace functionality integrated at testbed scale, this work takes the methodological aspect of developing IoT and cyber-physical systems to the next level. The integration of native hardware-based debug and real-time tracing into every observer node allows full testbed-wide access to the ARM real-time debug and trace collection infrastructure (CoreSight [12, 15]) at the program execution level. Access is provided remotely in exactly the same manner as on a single developer's desk to all devices-under-test. This alleviates the need for inflexible instrumentation of the software code run on a DUT as well as invasive run-stop debugging. In addition, this testbed integrates high fidelity power

profiling at nA resolution, dynamic control of the power supply and highest precision tracing and actuation of a set of DUTs based on Global Navigation Satellite System (GNSS) time synchronization. The testbed features a well-defined and open interface for test creation and test result fetching. A Python-based library and command line tool provides support for automated test management and visualization. This paper contains the following contributions:

- It proposes a testbed architecture that combines state-of-the-art debug and trace capabilities with accurate high-dynamic range measurements and actuation.
- Characterization of the system implementation.
- Demonstration of capabilities of FlockLab 2 in a case study.
- Open-source hardware design and software source code.

Sec. 2 gives an overview of the testbed landscape. In Sec. 3, we discuss the design of FlockLab 2 and characterize its implementation. In Sec. 4 we demonstrate the capabilities of FlockLab 2.

## 2 PAST EXPERIENCE AND RELATED WORK

The design of FlockLab 2 is heavily influenced by 8+ years of experience in developing and operating the FlockLab 1 testbed [8]. This testbed was based on the very successful target-observer model [8, 11] with multi-modal capabilities to monitor and influence devices-under-test at very high precision and fine-grained resolution. The FlockLab 1 testbed has been operated publicly since 2012. It ran over 70'000 tests by more than 370 users from more than 130 institutions in 30 countries. In addition, the testbed has been used by students in hands-on courses and many student projects.

Over time, a number of extensions based on the original concept of FlockLab 1 have been implemented [9, 10] calling for a revisit of the original concept with improved performance figures. Existing competitor testbeds each provide interesting features. However, none of them combine all three capabilities: (1) in-situ debug and trace, (2) high-dynamic range power profiling, and (3) accurate timing. In the following, we give an overview of the current testbed landscape.

TWIST [6] and Indriya2 [2] are both based on USB interconnects. Therefore they do not provide elaborate debug and trace features or accurate observations of hardware behavior like precise timing or power. On TWIST the power supply can be controlled by turning the USB interface to the targets on or off. Furthermore, a hierarchical back-channel using USB and Ethernet allows scalability.

The D-Cube [14] testbed focuses on benchmarking wireless protocols in pre-defined scenarios with a technique to embed test parameters directly in the software for the DUT, control RF interference and the automated publication of test metrics. In addition to serial logging, it supports setting and tracing GPIO pins and allows power consumption measurements. However, it does not support the use of debug and trace capabilities of modern MCUs.

FIT IoT-Lab [1] supports a wide range of sensor nodes (MSP430 to ARM Cortex-M8) at many different locations. Basic debug and trace based on JTAG and monitoring of power consumption is supported. Furthermore, the testbed supports injecting and sniffing radio packets and monitoring on a single frequency RF channel. To the best of our knowledge, it does not support accurate timing for control and measurements.

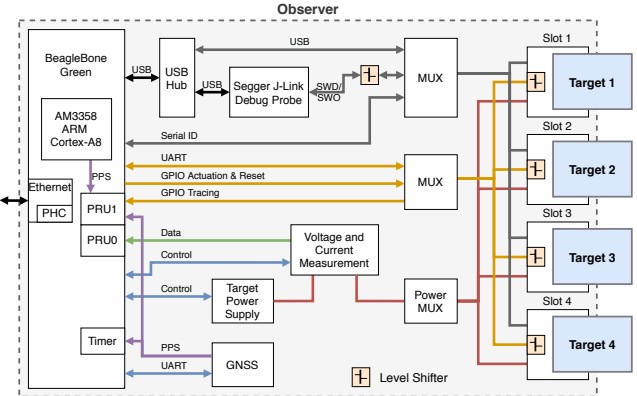

**Figure 2: FlockLab 2 observer architecture.**

Shepherd [5] focuses on recording and replaying energy harvesting power traces for research in batteryless IoT devices. The architecture supports basic debugging and GPIO tracing. Power measurements are supported up to 50 mA which is limiting for modern long-range radios with high transmit power. Currently, there is no publicly available instance of the Shepherd testbed.

## 3 A REAL-TIME TRACING ARCHITECTURE

An IoT testbed needs to support multi-modal distributed interaction and tracing. We identify the following key requirements for a state-of-the-art testbed for wireless IoT devices:

- Support for native debug and trace infrastructure.
- Accurate and high-dynamic range power measurements (sub-µA sleep current up to radio TX current of 170 mA).
- High-precision timing (sub-µs accuracy) across the distributed testbed.

The FlockLab 2 testbed architecture consists of a testbed server hosting data services and the web interface, a set of distributed observers carrying the instrumentation and providing connectivity and the devices-under-test (DUTs) also termed the *target devices*. In FlockLab 2, multiple targets, typically manifested by different sensor node architectures, are supported on each observer system. Each target device is connected to the observer hardware using a multiplexer crossbar allowing a user to select a distinct target hardware architecture without physical intervention (see Fig. 2). Using this multiplexing allows to run tests on different target device architectures physically collocated, e.g. to compare different radio architectures side-by-side. The independent and stateful observer, which stores tracing data locally, allows for a strong coupling between observer and target (see Fig. 3). This enables highest accuracy and throughput of the DUT instrumentation especially when comparing to direct out-of band back-channels of early testbed architectures [17].

The basic services of FlockLab 1 are continued: actuation and tracing of serial port and GPIO pins, target programming, power tracing and adjustable target supply voltage. The new system supports a native in-situ debug and trace service and generally a higher fidelity of the aforementioned basic services as well as an extended

testbed layout covering also wide-area distances [16]. The instrumentation for measuring the power consumption has been significantly improved: nA current measurements resolution, peak power up to 500 mA, a sampling rate of 64 kHz, electrical isolation of target devices to not perturb low-power measurements.

## 3.1 Observer Instrumentation Platform

Each observer consists of a Linux host system, a main board and several target adapter boards hosting up to four different target devices. The four target slots are connected using a multiplexing unit that routes all signals on the observer main board.

*3.1.1 Linux Single-Board Computer as Observer Host Platform.* A standard Linux single-board computer (SBC), a BeagleBone Green, is used as host platform for the decentralized stateful observers. All tracing data is recorded on the observer in order to alleviate the inherent bottleneck to the testbed server. The BeagleBone Green SBC includes a single core ARM Cortex-A8 processor and two single-cycle Programmable Real-Time Unit (PRU) co-processors for low latency tracing. It further features an Ethernet interface with integrated hardware support for network-based time synchronization (NTP/PTP), generic IO extensions and local Flash memory.

*3.1.2 Embedded Debug and Trace Integration.* The key feature on the FlockLab 2 observer is an integrated Segger J-Link OB debug probe. This gives native access to state-of-the-art ARM Cortex-M CoreSight debug and trace facilities which are built into modern systems-on-chip (SoC) [12]. This allows to utilize simple halting debug mode (where architectural state can be observed), single step execution, breakpoint units and Performance Monitoring Units (PMUs). CoreSight further provides an Embedded Cross Trigger mechanism to synchronize or distribute debug requests and profiling information across the SoC. Embedded Trace Macrocells (ETM trace unit) or Program Trace Macrocells (PTM trace unit) allow to trace program execution at runtime and without instrumentation in the code that (i) alters program behavior and (ii) needs to be adapted for every single analysis step. The trace macrocells can either be captured using an on-chip trace buffer or accessed via the generic Serial Wire Debug (SWD) connection implemented on the J-Link debug probe acting as off-chip trace port analyzer (see Fig. 3). Dedicated synchronization points and global 64-bit timestamps across the whole SoC architecture can be enabled in the tracing architecture to gain accurate temporal context of an application and its interaction with the underlying hardware at runtime. The debug and tracing architecture is implementation specific and can be found in the respective microprocessor documentation.

The use of SWD on FlockLab 2 with the SWDCLK and SWDIO signals as well as a dedicated Serial Wire Output (SWO) trace port is a tradeoff between bandwidth and pin count. By using data buffer exchange capabilities of the debug probe, e.g. with Segger Real Time Transfer (RTT) software technology that can be easily integrated with user code, high-speed and little impact data transfer from the target to the observer is supported. For example, this allows more efficient `printf()`-style logging on the target.

Besides the native hardware support for debug and trace, the main advantage lies in the ability to connect to all standard developer tools allowing interactive debug sessions on the testbed

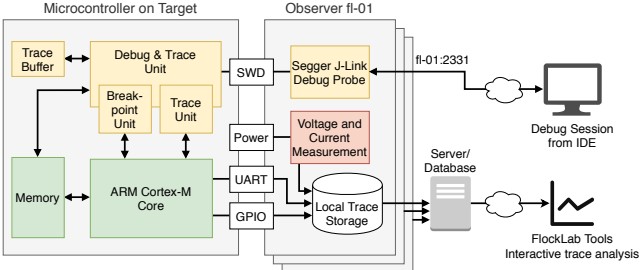

**Figure 3: Tracing instrumentation based on the in-situ debug and trace unit and external capabilities.**

directly from a developer's IDE or use ready made tooling for automating tasks. A lot of time in sensor networks and IoT research has been spent in developing custom tooling and a host of scripts incompatible with industry standard debugging and profiling tools.

*3.1.3 Power Tracing.* Highly accurate and high-dynamic range power profiling is based on the RocketLogger embedded measurement device [13]. It combines two measurement principles using seamless autoranging: a shunt ammeter (high current range), as well as a feedback ammeter (low current range) provide the necessary precision and range to measure sleep currents (sub-μA and peak power consumption (100's of mA) of modern radios. The measurement circuit allows to measure the current through as well as the voltage at the target. A careful electrical isolation of all target IO lines allows to accurately measure on the order of nAs for the lowest power modes. Measurements are performed by precision analog-to-digital converters (ADCs) and periodically transferred to the single-board computer's storage via PRU0. To compensate for hardware and manufacturing variations, the voltage and current measurement circuit on each observer is calibrated before first use.

*3.1.4 Serial Logging and Forwarding.* Serial port communication via UART or USB is supported on each observer. This allows logging of simple `printf()` based console output and supports interactive communication with the target during tests by forwarding the serial port via TCP to the testbed user. To achieve high performance and accurate timing, logging is implemented in C and events are timestamped using the GNSS or PTP disciplined system clock.

*3.1.5 Logic Actuation and Tracing.* On each observer, 5 target GPIO pins can be captured and 2 target GPIO pins can be actuated. In FlockLab 2, logic tracing is implemented on the programmable real-time unit PRU1, which allows to acquire highly accurate timing trace data. Logic actuation is implemented by a Linux kernel module and the actuation events are logged by the PRU based logic tracing as well.

*3.1.6 Testbed-wide Time Synchronization.* Accurate timing across all signals for tracing and actuation on a single observer platform as well as across the whole testbed is one of the most important success factors of a distributed IoT testbed. With recent advances in higher system clock rates and ever more timing critical behavior in advanced communication schemes [10] the requirements for accurate timing is in the sub-microsecond scale. Since this testbed is designated to support long-range communication where observers

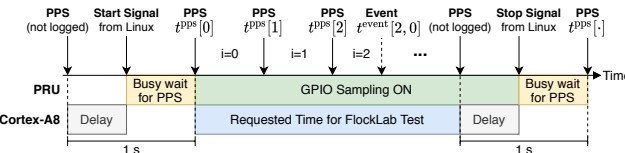

**Figure 4: Logging of PPS signal alongside the target signals for accurate time synchronization.**

might be distributed over kilometers, synchronization needs to work independent of other observers and their location.

The local Linux system time referenced to UTC and disciplined by GNSS serves as time reference for all testbed services. The integrated GNSS receiver (u-blox M8) generates an accurate pulse-per-second (PPS) signal which is tracked in dedicated hardware timers on the single-board computer. For accurate timing of the logic tracing, the PPS pulse is logged alongside the target signals (see Fig. 4). A linear correction factor is calculated for each epoch $i$ and applied to timestamp of the logic tracing event (numbered by $k$) once a test has completed.

$$t_{\text{Global}}^{\text{event}}[i, k] = t_{\text{PRU}}^{\text{event}}[i, k] \cdot \frac{t_{\text{Global}}^{\text{pps}}[i+1] - t_{\text{Global}}^{\text{pps}}[i]}{t_{\text{PRU}}^{\text{pps}}[i+1] - t_{\text{PRU}}^{\text{pps}}[i]}$$

Observers at locations with limited GNSS signal reception can use the Precision Time Protocol (PTP) as a fallback solution to discipline the Linux system clock. A prerequisite for this is a network infrastructure which fulfills the requirements of the PTP protocol. The PPS signal required for accurate logic tracing is based on Linux system time generated by a kernel module. Using hardware assisted timestamping at the PHY and MAC layer of the single-board computer's Ethernet interface allows to achieve synchronization accuracy in the order of ∼1 µs for PTP [7] compared to ∼50 ns for GNSS [10].

For the debug probe, incoming messages from the SWO trace port are timestamped using system time. The debug unit can be configured to export local timestamps via SWO. These can be converted to system time by applying a piece-wise linear regression similar to the correction factor for logic tracing described earlier.

*3.1.7 Target Adapter.* The target adapter is mainly a hardware adapter bridging form-factor and pinout. It may contain configuration options (e.g. jumpers) and extra debug pins depending on the target platform. Additionally, it contains a serial ID chip for automated identification of every target connected to an observer.

*3.1.8 Power Generation and Reset.* The target supply voltage is generated using an low-dropout (LDO) regulator controlled by a digital-to-analog converter (DAC) based reference voltage. This allows to dynamically control the target supply voltage. The target can be reset either by controlling the reset pin or by a full power-off-reset (POR). These power and reset capabilities enable to test under different operating conditions and under most realistic conditions.

*3.1.9 Target Programming.* Programming of the microcontroller of the target devices is performed either using a bootloader (BSL) or native single wire debug (SWD) for ARM based devices.

| Target | MCU | Arch. | Radio |
|---|---|---|---|
| Tmote Sky / TelosB | MSP430F1611 | MSP430 | CC2420, 802.15.4, 2.4 GHz |
| DPP2 CC430 [3] | CC430F5147 | MSP430 | CC430 SoC, CC1101-based, 868 MHz |
| DPP2 LoRa [3] | STM32L4 | ARM M4 | SX1262, LoRa/FSK, 868 MHz |
| nRF52840 Dongle | nRF52840 | ARM M4 | nRF52 SoC, 802.15.4/BLE, 2.4 GHz |

**Table 1: Target devices supported on FlockLab 2.**

## 3.2 Testbed Management and User Interface

*3.2.1 Testbed Infrastructure.* The testbed is orchestrated by a server which executes the scheduled tests, provides a MySQL database, provides storage space for test results, hosts the web interface and exposes an API for automated test scheduling and fetching. The database stores test scheduling information and configuration as well as the current state of the hardware infrastructure (e.g. which target is connected to which observer).

*3.2.2 API and Visualization.* FlockLab 2 focuses on fully autonomous test execution but also supports live interactions. Tests are configured and scheduled using a single XML file. This test configuration file allows to (1) select the target platform and the testbed nodes, (2) enable and configure actuation and tracing services, and (3) include one or more program images which will then be flashed to the targets. Real-time interaction during test execution is supported via the serial communication service (read/write) and via a remote debug session using the integrated Segger J-Link debug probe which allows to set breakpoints, halt the execution, read processor state and to retrieve data via SWO.

The results are stored on a server and can be downloaded as an archive file. Test management (creating and stopping tests, retrieving status information, downloading results) as well as an intuitive visualization of results is supported via web interface or the `flocklab-tools` command-line tool executed on the user's computer.

## 3.3 Publicly Available Testbed

The FlockLab 2 testbed is implemented as a public service with currently 15 active observers (Fig. 1). 12 observers are distributed on the floor of an office building and 3 observers are installed at remote long-distance rooftop locations [16]. Additional observer hardware will extend the testbed to 30+ nodes. The testbed can be publicly accessed via the website[1]. where we also publish documentation, examples, the hardware design and software source code.

Currently, the four target platforms listed in Tab. 1 are available. Additional target platforms can easily be added thanks to the generic target interface.

## 3.4 FlockLab 2 Observer Key Characteristics

We provide a characterization of the FlockLab 2 observer in Tab. 2.

## 4 USING FLOCKLAB 2 IN PRACTICE

In order to demonstrate the capabilities of FlockLab 2 regarding features and performance, we discuss an example of network flooding based on synchronous transmissions [4] on a FSK/LoRa radio [3]. Building and scaling-up a communication protocol based on synchronous transmissions requires a very careful arbitration of all

---

[1]https://www.flocklab.ethz.ch

| Target Power Supply | |
|---|---|
| Voltage range | 1.1 - 3.6 V |
| Voltage resolution | 13.5 mV |
| Max. current | 500 mA |
| **Logic Actuation** | |
| Timing accuracy | <100 μs (typ.) |
| **Power Tracing** (see [13] for details) | |
| Max resolution / sampling rate | 15.625 μs / 64 kHz |
| Voltage accuracy | 0.37% + 4 mV |
| Current accuracy (low range 0 - 2 mA) | 0.01% + 60 nA |
| Current accuracy (high range 2 - 500 mA) | 0.02% + 48 μA |
| **Logic Tracing** | |
| Max resolution / sampling rate | 0.1 μs / 10 MHz |
| Max burst event rate (≤ 2000 edges) | 10 MHz |
| Max continuous event rate (typ.) | 900 kHz |
| Timing Accuracy (GNSS) | <0.25 μs (typ.) |
| **Serial Tracing** | |
| Max continuous throughput | 460 kbaud |
| Timing accuracy | <10 ms (typ.) |

**Table 2: Characterization of the FlockLab 2 observer.**

radio activities and extensive debugging as transmissions and corresponding interrupts need to be correctly aligned. It is therefore a suitable example to showcase the capabilities of FlockLab 2. In this example, we use the DPP2 LoRa target which consists of an STM32L433 Cortex-M4 microcontroller and a Semtech SX1262 radio [3] with a 0.28 μA standby current and 7 μs wake-up latency.

## 4.1 Simple Synchronous Transmission Protocol

Gloria is an optimized multi-hop network flooding protocol based on Glossy [4]. As depicted in Fig. 5, all nodes synchronously retransmit a received message a pre-defined number of times (3 times in our example) in subsequent transmission slots. Contrary to Glossy, Gloria nodes listen to a message only once. For the setting used in this example, the relevant values are InitOverhead = 1.783 ms and SlotTime = 4.548 ms (see Fig. 5). Both values are fixed for each specific radio configuration (in this case FSK 250 kbit/s) and have been determined using the datasheet and measurements.

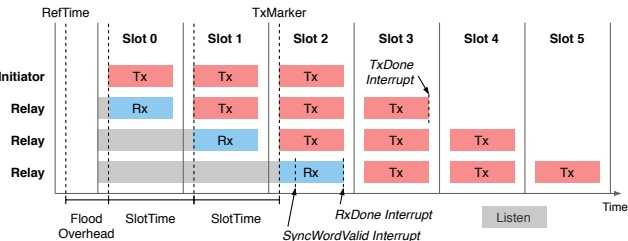

**Figure 5: Gloria floods use concurrent re-transmission.**

## 4.2 Testing Workflow

*4.2.1 Creating a Test.* First, the software for the target platform is compiled using the standard toolchain/IDE. Then, an XML test configuration file is created containing the nodes, images and platform to be used, test duration, actuation, tracing and debugging services

```xml
<testConf xmlns="http://www.flocklab.ethz.ch">
    <generalConf>
        <name>FlockLab XML template</name>
        <schedule><duration>60</duration></schedule>
    </generalConf>
    <targetConf>
        <obsIds>2 4 6 7 9</obsIds>
        <voltage>3.3</voltage>
        <embeddedImageId>Image_1</embeddedImageId>
    </targetConf>
    <serialConf>
        <obsIds>2 4 6 7 9</obsIds>
        <baudrate>115200</baudrate>
    </serialConf>
    <powerProfilingConf>
        <obsIds>2 4</obsIds>
        <samplingRate>1000</samplingRate>
    </powerProfilingConf>
</testConf>
```

**Listing 1: FlockLab 2 test configuration example.**

configuration. A minimalist example is shown in Listing 1. This is then uploaded to the FlockLab 2 server using the web interface or the flocklab-tools. On the server, the test is then scheduled. The server initiates the start of the test at the time specified, distributes the target images and configures all testbed services.

*4.2.2 Interaction During Test Execution.* Progress is monitored on the web interface where information on configuration and status is available. If the serial forwarding service is used it is possible to connect to an individual observer for the duration of the test using a TCP connection, e.g. by using netcat. Likewise an interactive debug session can be opened from the IDE on the user's computer to a GDB debug server running on the observer.

*4.2.3 Analyzing Test Results.* After the test completes, the server fetches all results from the observers and combines them into a single test result archive file that can be used for custom post-processing or can be visualized using the flocklab-tools (an example is depicted in Fig. 7).

## 4.3 Debugging and Analysis of the Protocol

*4.3.1 Embedded Debugging at Testbed Scale.* In this example, 8 nodes are performing a Gloria network flood. To validate the correct protocol implementation, we use the debugger functionality which allows to extract internal variables. Concretely, we want to verify the correct calculation of the time of the next transmission (TxMarker in Fig. 5). In Figure 6, for each node, radio activity is shown in the first row (orange bars), the radio interrupts are shown in the second row (black bars). For node 4, the power trace (black line) is shown as well. A breakpoint has been set on the first TxDone interrupt of node 9 that can be inspected using a remote debug session to the target (see Sec. 3.1.2). Since the breakpoint halts this specific microprocessor, the captured traces show no more GPIO events after the node reached the breakpoint. The variables inspected at the breakpoint show e.g. slot_index = 2; message_size = 30 as expected. The variables reconstructed_marker and current_tx_marker correspond to RefTime and TxMarker in Fig. 5, respectively. The extracted time difference value of 10.879 ms conforms with the expected value for

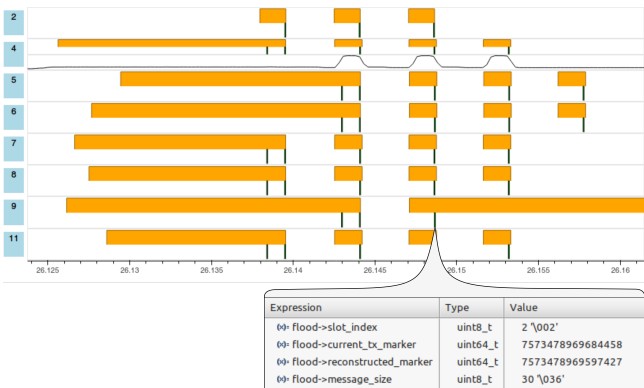

**Figure 6: With the debug service, a breakpoint is set on the first TxDone interrupt on node 9. This allows to extract the values of the internal variables at that point in time.**

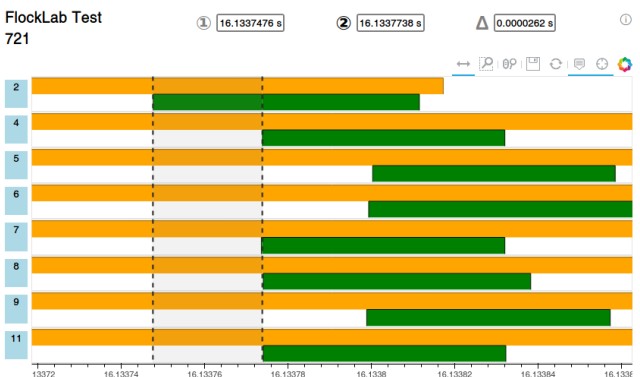

**Figure 7: The transmissions (TxDone interrupts; green bars) of different nodes are not aligned because an offset timing parameter is not set correctly. The logic tracing service allows to detect and correct this erroneous behavior at interrupt-level granularity.**

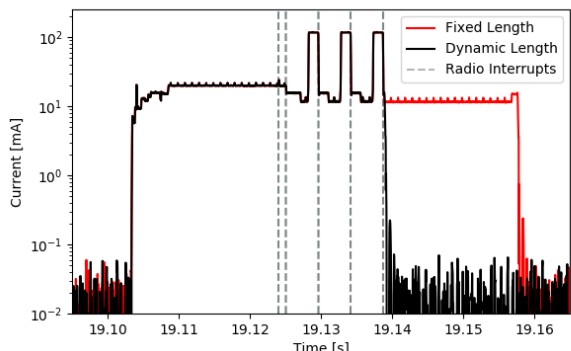

**Figure 8: High-dynamic range power tracing is used to validate and optimize low-power behavior.**

*4.3.3 Optimizing for Low Power Consumption.* To maximize the lifetime of a battery-powered cyber-physical system, careful optimization and orchestration of the low-power modes is required. In this example, we validate the low-power behavior of the Gloria flood implementation by using the power tracing service (Sec. 3.1.3) together with the logic tracing capabilities (Sec. 3.1.5). In a simple implementation communication is executed in a fixed-length active window (see Fig. 8). In the optimized case a node will transit to low-power sleep mode immediately after completing a required action, e.g. sending and receiving data. The difference between fixed-length and dynamic active window sizes can be seen in Fig. 8 together with the respective radio interrupts.

## 5 CONCLUSIONS

In this paper, we present the second-generation testbed FlockLab 2 which combines industry standard debug and trace support with accurate high-dynamic range power and timing measurements. Relevant design aspects including the distributed testbed-wide time synchronization and the in-situ debug and logging capabilities have been demonstrated with real-world applications. These aspects make the testbed a valuable tool for developing and benchmarking distributed IoT systems.

FloodOverhead+2·SlotTime. This confirms the correct calculation of TxMarker in the synchronous Gloria flood.

*4.3.2 Timing Validation using Logic Tracing.* In synchronous protocols, transmissions and corresponding interrupts need to be correctly aligned. This is traditionally done by instrumenting code and tracing GPIO pins with the logic tracing service (see Sec. 3.1.5). In Fig. 6, radio activity with two interrupts (SyncWordValid and RxDone) corresponds to a message reception and radio activity with a single interrupt (TxDone) corresponds to a transmission. Re-transmissions are scheduled based on the timing of received messages. For this, the SyncWordValid timestamp is used to calculate individual start times on each node. For this, the exact time offset between the start of the transmission and the SyncWordValid interrupt needs to be calibrated. In the example in Fig. 7, this offset is not set correctly and consequently synchronous transmissions are not aligned. Using logic tracing, this malfunction can be detected (green lower bars) and the correct value can be determined.

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
