# OpenReview forum: "FlockLab 2: Multi-Modal Testing and Validation for Wireless IoT"
_sigmobile.org/MobiCom/2020/Workshop/CPS-IoTBench — CPS-IoTBench 2020_

### Official Review · AnonReviewer1 · 2020-06-10
**Nice paper, very valuable contribution to the community**

**Rating:** 9
**Confidence:** 5

**Review:**

The paper is in a very good shape.
The contribution is highly relevant for the community and of excellent quality; keep this line of work going! The paper is very well described, it is original and highly significant.
Pros: very interesting platform, open to everyone, lots of work went into this and helps the community
Cons: the number of references to own papers is rather high while related work is only briefly discussed

---

### Official Review · AnonReviewer2 · 2020-06-26
**Strong accept**

**Rating:** 9
**Confidence:** 5

**Review:**

This paper outlines FlockLab2, specifically the extensions that allow for remote, distributed debugging. The paper outlines the challenges of the extensions, specifically in time synchronization as well as highly accurate power tracing.  It also steps through an example of how the debugging works for a complex, glossy-based protocol.

Strengths:
+ well written, easy to read
+ important testbed extensions to further extend the use of testbeds for debugging and understanding new systems

Weaknesses:
- very implementation oriented and light on "research"

Despite the implementation oriented nature of the paper, it is highly relevant to the workshop topics, and dissemination of the existence of these extensions to the testbed infrastructure is important and valid.

---

### Official Review · AnonReviewer3 · 2020-06-27
**A well-written manuscript about an equally well-engineered testbed**

**Rating:** 9
**Confidence:** 5

**Review:**

The paper presents V2 of FlockLab. The V2 significantly enhances(replaces) the earlier system and provides some unique features not presented in the competing testbeds. It is a well-engineered system that, among other things, notably exploits the state of the art debugging and tracing functions of the latest MCUs. I can already see that this would be very useful for the community. Definitely a strong accept.

The case study, an implementation of a variant of Glossy on a Sub-GHz radio seems to be a good test for precise timing functionality provided in the testbed. However, not sure why the authors rebranded a previously known variant of Glossy with yet another name, Gloria ( without citing documents that first proposed it).

Pros: A well-engineered system. The manuscript is easy to follow. The paper clearly differentiates its contribution from the earlier testbeds by providing a few new unique features
Cons: Many self-citations. No citations to earlier appearances of "Gloria".

---

### Official Review · AnonReviewer4 · 2020-06-30
**A useful testbed design and built that is useful for low power wireless testing**

**Rating:** 9
**Confidence:** 5

**Review:**

This paper presents Flocklab 2. Flocklab has three main
capabilities. Native debug/tracing. Power measurement. Precise
timestamping. Flocklab supports several IoT development platforms
which can be observed by the host computer, which is a BeagleBone
Green device.

The testbed is design and built and deployed to some extent. Thus, the
description tells us how to build such testbeds in practice. In terms
of capabilities, such capability has existed at the node level, and
also the prior version, FlockLab, included some
capabilities. Additional interfaces and capabilities (e.g., serial
debugging) from FlockLab are incorporated in the new version. Thus,
this work represents a significant evolution.

The authors do not articulate clearly the scenarios in which such a
testbed is useful. The testbed is likely an overkill if someone wants
to test an L3 network protocol. Every testbed has its best use case
and not so great use case.

A useful future work could be to discuss how to utilize all the
information collected. Oftentimes, it is much easier to collect the
type of fine-grained information than utilize it effectively.

The authors are encouraged to improve the caption for fig 6/7 so that
it is self-contained and understandable with proper context.

Overall, this paper is a solid example of building a testbed to
evaluate wireless devices and protocols.

---

### Decision · Program_Chairs · 2020-07-07

Accept